# Non-invasive prenatal paternity testing by analysis of Y-chromosome mini-STR haplotype using next-generation sequencing

**Wenqian Song[1], Nan Xiao[1], Shihang Zhou[1], Weijian Yu[1], Ni Wang[1], Linnan Shao[1], Ying Duan[1], Mei Chen[1], Lingzi Pan[1], Yuexin Xia[1], Li Zhang[1], Ming Liu[2]***

**1** Dalian Blood Centre, Dalian, China, **2** Institute of Cytobiology, Dalian Medical University, Dalian, China

* liuminglinxi@163.com

**Data Availability Statement:** All relevant data are within the paper and its Supporting information files.

## Abstract

### Objectives

To assess the efficacy of Y-chromosome mini-STR-based next-generation sequencing (NGS) for non-invasive prenatal paternity testing (NIPPT).

### Methods

DNA was extracted from the plasma of 24 pregnant women, and cell-free fetal DNA (cffDNA) haplotyping was performed at 12 Y-chromosome mini-STR loci using the Illumina NextSeq 500 system. The cffDNA haplotype was validated by the paternal haplotype. Subsequtllly, the paternity testing parameters were attributed to each case quantitatively.

### Results

The biological relationship between the alleged fathers and infants in all 24 family cases were confirmed by capillary electrophoresis (CE). The Y-chromosome mini-STR haplotypes of all 14 male cffDNA were obtained by NGS without any missing loci. The alleles of cffDNA and paternal genomic DNA were matched in 13 cases, and a mismatched allele was detected at the DYS393 locus in one case and considered as mutation. No allele was detected in the 10 female cffDNA. The combined paternity index (CPI) and probability of paternity calculation was based on 6 loci Y-haplotype distributions of a local population. The probability of paternity was 98.2699–99.8828% for the cases without mutation, and 14.8719% for the case harboring mutation.

### Conclusions

Our proof-of-concept study demonstrated that Y-chromosome mini-STR can be used for NGS-based NIPPT with high accuracy in real cases, and is a promising tool for familial searching, paternity exclusion and sex selection in forensic and medical applications.

**Funding:** WS received research support by a Dalian Municipal Youth Science and Technology Star grant (2017RQ169) from Dalian Municipal Science and Technology Bureau, Dalian, China (http://www.kjj.dl.gov.cn/). The funder had no role in study design, data collection and analysis, decision to publish, or preparation of the manuscript.

**Competing interests:** The authors have declared that no competing interests exist.

## Introduction

The conventional prenatal paternity testing methods rely on invasive amniocentesis or chorionic villus sampling, which may lead to pregnancy-related complications and increase the procedure-related risk of miscarriage to 0.35% [1, 2]. Lo et al. [3] first isolated cell-free fetal DNA (cffDNA) from maternal plasma in the 1990s, which paved the way for developing non-invasive prenatal testing techniques [4]. Currently, non-invasive prenatal testing is widely used for detecting the rhesus D blood type [5], fetal aneuploidy like Down syndrome [6], fetal sex [7] and paternity [8]. Approximately 99% of the cffDNA is shorter than 313 base pairs (bp), and is largely derived from trophoblast destruction, or the apoptosis of fetal hematopoietic cells and trans-placental transfer to a lesser extent [9, 10]. A previous study showed that the circulating cffDNA had a mean half-life of 16.3 min and was undetectable in the maternal plasma 2 h post-delivery. This indicates that cffDNA testing is not affected by carryover from previous pregnancies [11].

In forensic science, Y-STR typing is used as an additional tool for confirming the paternity or identifying individual male sequences in complex DNA mixtures along with autosomal STR typing [12, 13]. Due to its uniparental inheritance, the results of Y-STR typing are interpreted on the basis of haplotype distribution, which reveals the familial relationship pattern [14]. A major challenge in non-invasive prenatal paternity testing (NIPPT) is the detection of fetal-specific markers from the paternal genetic material in the maternal plasma. Therefore, autosomal STR-based prenatal paternity testing is not used frequently due to maternal DNA contamination and the requirement of short target sequences. Since Y-specific cffDNA can be easily distinguished from the abundant maternal DNA signals in the maternal plasma, detection of Y-chromosome markers can be used to assess the paternity of male fetuses by capillary electrophoresis (CE) or SNP-based next-generation sequencing (NGS) [15, 16]. In addition, mini-STR loci detection is an effective strategy for recovering genetic information from highly degraded DNA samples, and was solely used to fingerprint 20% of the degraded DNA samples from the aftermath of the 9/11 World Trade Centre terrorist attacks with reduced PCR amplicon sizes [17, 18]. The aim of our study was to evaluate the efficacy of Y-chromosome mini-STR-based NGS for NIPPT, and quantify the extent of match using paternity testing parameters.

## Materials and methods

### Sample collection

Peripheral blood samples were collected from 24 pregnant women undergoing prenatal tests, and their male partners, at the Dalian Blood Centre from April 2018 to December 2019. The age of the women ranged from 26–38 years, and the gestational age ranged from 21–37 weeks (17 women in the second trimester and 7 in the third trimester). Buccal swabs were collected from the infants after delivery. All participants signed the informed consent, and the study was approved by the Dalian Blood Centre Ethics Committee.

### DNA extraction

Around 5 mL peripheral blood was collected from the pregnant women in anticoagulant-treated tubes. Plasma was isolated from the whole blood sample using two-step centrifugation at 1500 ×g for 10 min and 13,000 ×g for 10 min [19]. The supernatant was collected and stored at -80˚C. Plasma DNA was extracted from 2 mL cell-free maternal plasma using the MagPure Circulating DNA Maxi Kit (Angen Biotech, Guangzhou, China) and eluted with 60 μL water according to the manufacturer's instructions. The 2800M Control DNA (Promega, Madison, WI, USA) was used as the positive control, and nuclease-free water was used as the PCR blank.

Genomic DNA was extracted from the peripheral blood of the pregnant women and their husbands, and from the buccal swabs of infants using HiPure Tissue & Blood DNA Kit (Angen Biotech). The quantity and purity of the 1 µL extracted DNA were determined by sodium dodecyl-sulphate polyacrylamide gel electrophoresis (SDS-PAGE) with Tanon 1600 Gel Imaging System (Shanghai Tanon Science & Technology, Shanghai, China) and NanoDrop 2000 spectrophotometry (Thermo Fisher Scientific, Waltham, MA, USA) respectively. Presence of contaminants such as RNA and proteins can increase the absorbance of the DNA samples at 260, resulting in overestimation of DNA concentration. Therefore, spectrophotometric measurement of DNA was supplemented with SDS-PAGE to avoid interference from these contaminants. The DNA concentration of each sample was estimated on the basis of a molecular weight marker band.

## Paternity testing by Capillary Electrophoresis (CE)

Paternity testing was performed using the Microreader™ 21 ID System (Microread Genetics, Beijing, China) and Microreader™ 29Y ID System (Microread Genetics, Beijing, China). The genomic DNA extracted from all subjects was amplified using Life ECO Thermal Cycler (Bioer Technology, Hangzhou, China) according to the manufacturer's instructions. Autosomal STR genotyping was performed using the 3130 Genetic Analyzer system (Thermo Fisher Scientific). The genotyping results were analysed using GeneMapperTM v3.0 software (Thermo Fisher Scientific).

## Library preparation and NGS

One microliter DNA extract from cell-free maternal plasma was amplified using the barcoding primers of the following 12 Y-chromosome mini-STR loci: DYS439, DYS437, DYS643, DYS393, DYS570, DYS392, DYS549, DYS460, DYS458, DYS576, DYS438 and DYS533 (S1 Table). The 2800M Control DNA was used to construct the library. The DNA library was prepared using KAPA Library Amplification Kit (Illumina, San Diego, CA, USA) according to the manufacturer's instructions. Specialized adapters and indexes were added to both ends of barcoding sequences (Fig 1). Targeted amplifications were performed in single-tube reactions on a VeritiTM 96-Well Thermal Cycler (Applied Biosystems, Foster City, CA, USA) with the following cycling conditions: 95 ˚C for 5 min; 35 cycles of 94 ˚C for 30 s, 55 ˚C for 4 min; 72 ˚C for 60 min. Sample indexing with specialized adapters was performed with the following parameters: 95 ˚C for 5 min; 15 cycles of 95 ˚C for 30 s, 60 ˚C for 30 s, 72 ˚C for 30 s; 72 ˚C for 5 min. The quantity and purity of the DNA libraries were assessed by SDS-PAGE. Subsequently, the adapter-ligated templates were purified using MagPure A3 XP beads (Angen Biotech). The libraries were quantified with KAPA Library Quant DNA Standards & Primer Premix Kit (Illumina) using the Qubit fluorometer (Thermo Fisher Scientific, Waltham, MA, USA). Each DNA library was normalized to 2nM and pooled in equal volumes. Finally, the DNA libraries were diluted to the concentration of 10pM and sequenced by paired-end 150 bp reads on the NextSeq 500 system (Illumina) (Fig 1).

## Data analysis

Quality control (QC) analysis of raw FASTQ data was performed using FastQC software. The FASTQ reads were mapped against reference sequences from GenBank to determine the mini-STR haplotypes of plasma DNA. The sequence and length variants per sample per locus were compiled and counted. The signal noise and stutter ratio of each locus was evaluated independently. The mini-STR haplotyping results of cffDNA were validated by that of paternal genomic DNA (Fig 2).

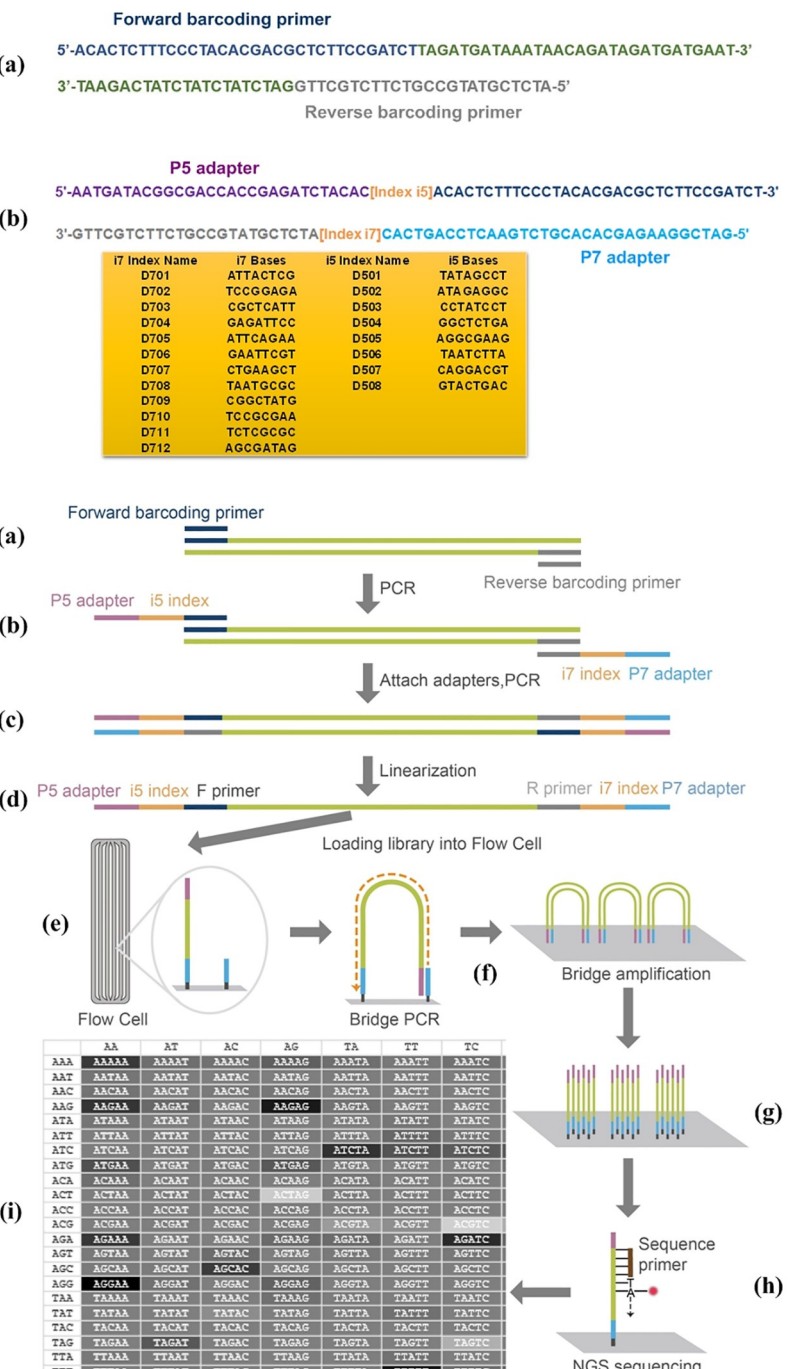

**Fig 1. Schematic representation of library preparation and NGS.** Data for D3S1358 is shown as a representative example. (a) Amplification of the target sequence with barcoding primers. (b) Second PCR amplification adding specialized adapters and indexes to both ends. (c) Purification of the amplicon and pooling multiple libraries together. (d) Linearization of DNA libraries. (e) Loading DNA libraries onto the flow cell. (f) Bridge PCR amplification. (g) Cluster generation. (h) Paired-end sequencing. (i) Base calling, alignment and data analysis.

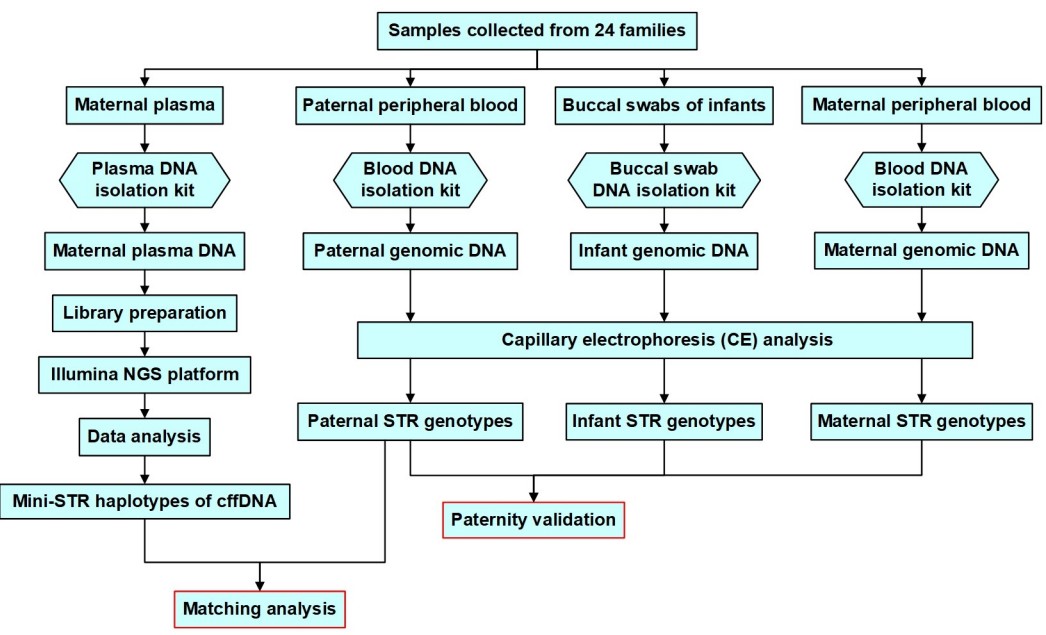

**Fig 2. Flow chart for experimental procedures and data analysis.**

## Calculation of paternity testing parameters

The combined paternity index (CPI) was calculated according to Eq (1) derived from Rolf's equation [20] where $fs$ is the frequency of the son's haplotype, $\mu$ is the mutation rate, n is the total number of Y-STR haplotypes, and m is the number of loci where the paternal and filial haplotypes differ. Probability of paternity was calculated by Eq (2) [21]. The CPI and probability of paternity rely on the haplotype distribution in the local population. The haplotype frequencies of the DYS392, DYS393, DYS438, DYS439, DYS437 and DYS458 loci were obtained from Guo's report [22] on the population genetics of Y-STR loci in the Liaoning population, which is the only local genetic information available at the Y-Chromosome STR Haplotype Reference Database (YHRD) (https://yhrd.org/). Given the limited information on the Y-STR loci haplotypes in the Liaoning population, only 6 out of the 12 loci tested in our study could be used for estimating haplotype frequencies. For example, one of 14 Y-haplotypes with 6 loci in our study matches two of the 838 Liaoning haplotypes available at the YHRD, that results in a haplotype frequency of 0.003521 ((1+2)/(14+838)).

Then, CPI and probability of paternity can be calculated based on this haplotype frequency. According to Eq (1), m = 0 and n = 1 were used in cases without mutation, CPI = $\frac{1}{fs}$. For the case with a mutation, the mutation rate $\mu$ can be calculated from YHRD. According to Eq (1), m = 1 was used, and CPI = $\frac{\frac{1}{2}\mu}{fs}$.

$$\text{If m} = 0, \text{CPI} = \prod_{a=1}^{n} \frac{1}{fs}$$

$$\text{If m} > 0, \text{CPI} = \prod_{b=1}^{m} \frac{\frac{1}{2}\mu}{fs} \; X \prod_{a \neq b}^{n} \frac{1}{fs} \tag{1}$$

$$\text{Probability of paternity} = \frac{CPI}{CPI + 1} \tag{2}$$

## Results

### Paternity validation

Paternity testing with mother, alleged father and infant was performed using commercial auto-somal STR genotyping kits by CE, which confirmed the biological relationship between the fathers and infants in all 24 family cases.

### STR haplotype-match analysis between cffDNA and paternal genomic DNA

The quantities of the 24 cell-free fetal DNA extracts are shown in S2 Table. Plasma DNA samples were sequenced on the Illumina NextSeq 500 platform, and the number of reads per sample per locus is shown in Fig 3. The average number of the sequence reads for each sample was 283535 (range 84732–819334). The target sequence length of all 12 mini-STR loci was less than 150 bp, of which only 3 STR loci were shorter than 150 bp as per CE (Fig 4). The Y-chromosome mini-STR haplotyping results of 14 male cffDNA samples were obtained without missing loci with NGS. The alleles of cffDNA and paternal genomic DNA were matched in 13 cases, whereas one mismatched allele (14→15) was detected at DYS393 in one case. Given the positive paternity identification, the mismatched allele at DYS393 was considered a mutation. No allele was detected in the 10 female cffDNA samples (S3 Table).

### Calculation of paternity testing parameters

The probability of paternity using cffDNA was 98.2699–99.8828% for the cases without mutation. The mutation rate of DYS393 is 0.00123 according to YHRD, and taking that into account, the CPI and probability of paternity for case No. 14 were 0.1747 and 14.8719% respectively (Table 1).

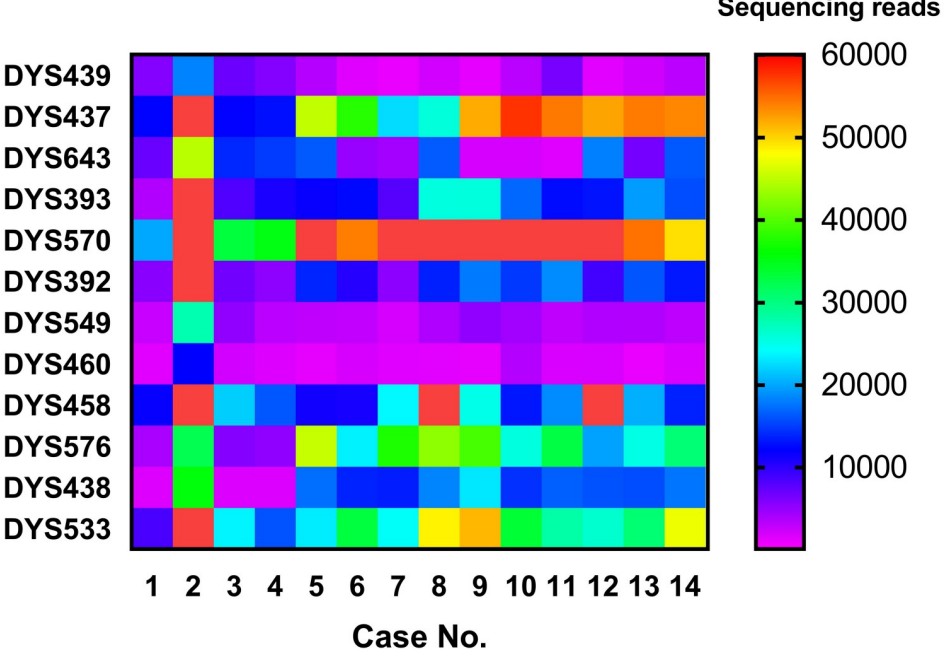

**Fig 3. The NGS sequencing reads per sample per locus of plasma DNA (range: 1068–140045 reads).**

## Discussion

Capillary Electrophoresis is the routine technique used for STR typing, and is widely used for individual and paternity tests in forensic investigations [23, 24]. NGS is a supplementary tool for forensic genetics owing to the large amount of genetic information that it can provide and process, high throughput function and low costs [25, 26]. Since the length of cffDNA in the maternal plasma is approximately 145-201bp, only short amplicons are available for fetal DNA analysis [9, 27]. Amplicons with similar length must be labelled with different fluorescent markers for CE, which limits the number of loci that can be multiplexed together. In contrast, NGS uses barcodes and index adapters that precludes the need for size separation between amplicons. As a result, numerous loci and multiple samples can be simultaneously analyzed in a single reaction. The mini-STRs were redesigned with primers flanking the repeat region to reduce the amplicon size for small DNA fragment detection [17]. Compared to the commercial kits using CE with amplicon length of 79–430 bp, the mini-STR amplicon lengths analyzed by NGS in our study were all less than 150 bp (Fig 4). Several reports have been published on SNP-based prenatal paternity testing combined with microarrays [28]. Since it relies on the differences in paternally versus maternally inherited SNP alleles, millions of SNPs have to be analyzed due to low polymorphism information content (PIC) leading to statistics with lower power. In addition, the routine use of SNPs in forensics is controversial since some SNPs may be present in the coding regions with bioethical implications stemming from confidentiality and privacy concerns since they may disclose sensitive information or are modified due to disease [29]. In contrast, STRs are widely used for routine forensic applications, and only a set of

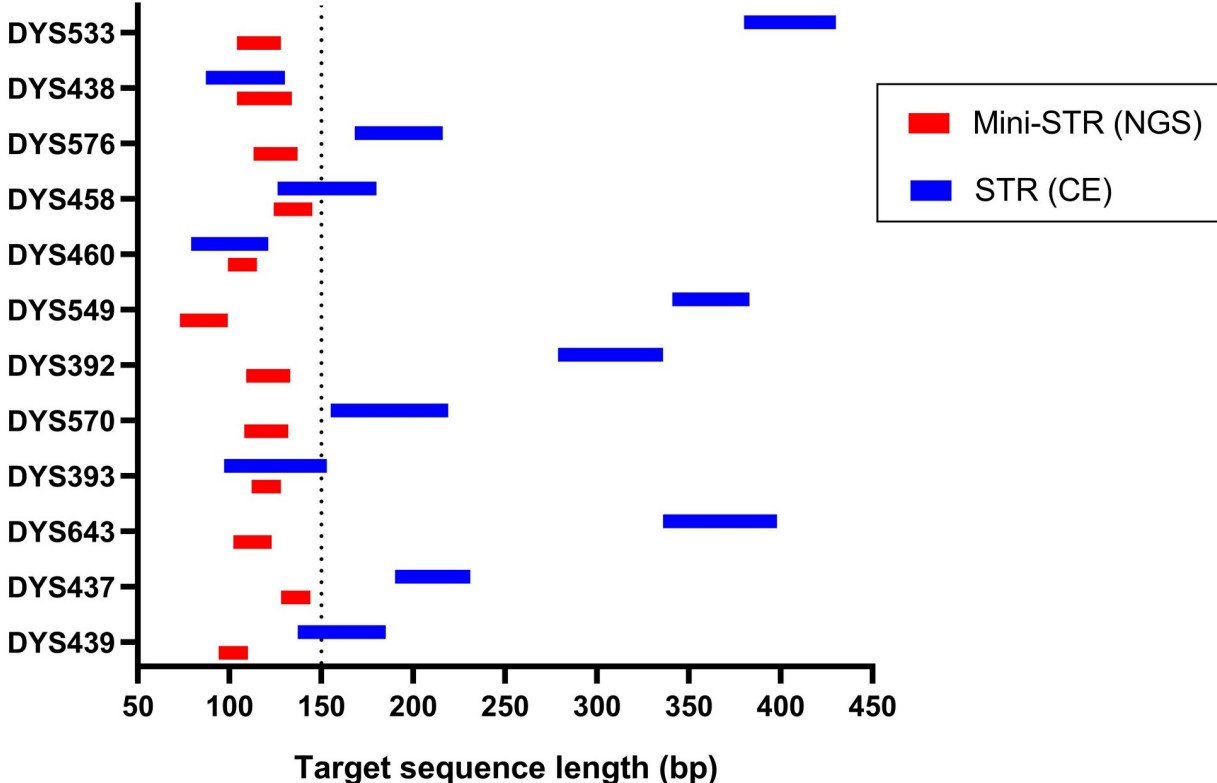

**Fig 4. Overview of target sequence length for all STR loci as per NGS and CE.** Target sequence indicates the PCR amplicon without barcoding and adapter primers.

**Table 1. The paternity testing parameters using cffDNA in the 14 cases with male fetuses.**

| Case No. | Haplotype frequency | CPI | Probability of paternity |
|---|---|---|---|
| 1 | 0.001174 | 852 | 99.8828% |
| 2 | 0.017606 | 56.8 | 98.2699% |
| 3 | 0.001174 | 852 | 99.8828% |
| 4 | 0.003521 | 284 | 99.6491% |
| 5 | 0.005869 | 170.4 | 99.4166% |
| 6 | 0.001174 | 852 | 99.8828% |
| 7 | 0.003521 | 284 | 99.6491% |
| 8 | 0.001174 | 852 | 99.8828% |
| 9 | 0.001174 | 852 | 99.8828% |
| 10 | 0.003521 | 284 | 99.6491% |
| 11 | 0.001174 | 852 | 99.8828% |
| 12 | 0.001174 | 852 | 99.8828% |
| 13 | 0.002347 | 426 | 99.7658% |
| 14[a] | 0.003521 | 0.1747 | 14.8719% |

[a]The case with a mutation at DYS393

12 STR loci is sufficient for paternity testing [30, 31]. Therefore, given the limitations of CE and SNP-based NGS, mini-STR-based NGS with high testing capacity and high PIC is a better choice for NIPPT. According to the guidelines of International Society for Forensic Genetics (ISFG), CPI and probability of paternity should be calculated in paternity testing [32, 33]. Due to strong linkage disequilibrium, the multiplying of single locus allele frequencies cannot be used for Y-STR CPI calculation. Instead, Y-STR-based paternity testing relies on complete haplotype frequencies [12]. In 7 of the 14 cases, the fetal haplotype were not detected in the 838 Liaoning haplotypes reported by Guo et al [22]. According to the ISFG algorithm [14], the haplotype frequency of these cases was 0.0012 (1/852), and the CPI and probability of paternity were 852 and 99.8828% respectively. Since most mutations in the STR loci are caused by DNA strand slippage during DNA replication, the Y-STR loci mutate independent of each other [12, 34]. In our study, one cffDNA sample had a mismatched allele (14→15) at DYS393, which was considered a mutation since paternity had been validated by CE. The sequencing result showed that the repetitive [AGAT] motif of fetal and infant allele had an extra repeat compared to the paternal allele. Unlike the autosomal loci, Y-STR haplotyping may lead to false-positive results if the alleged father and child belongs to the same paternal lineage [16]. For example, the fetus may have the same Y-chromosome as his biological father's brother or even grandfather's brother. In this case, Y-STR haplotyping is ineffective if the alleged father and biological father share the exact same Y-chromosome. Thus, Y-STR results must be confirmed with autosomal STR markers when the samples can be safely collected from the babies after birth or from aborted embryos. For this reason, Y-STR haplotyping is normally used along with autosomal STR genotyping to determine paternity. However, it is a viable option in forensics for identifying the paternal male relatives of an unknown perpetrator during large-scale voluntary DNA screening, whereas autosomal STR profiling can only trace the close relatives of the perpetrator [12]. In cases of sexual assault, the cffDNA from pregnant victims bearing male fetuses may provide valuable information regarding the DNA sequence of the perpetrator. Furthermore, for pregnant women with more than one sexual partner, NIPPT with Y-chromosome mini-STR can help exclude paternity if the male fetus and alleged father show no match. Although prenatal screening has been widely accepted, NIPPT is fraught with ethical

concerns [35]. It is at present illegal in China in order to avoid its misuse in determining sex of the fetus or in establishing paternity. Nevertheless, we recommend that NIPPT with Y-chromosome mini-STR should be used for unwanted pregnancy as the result of sexual assault or for sex selection in case of gene defect and inherited disease. Sex selection can be a double-edged sword, especially among the economically underprivileged population. While preference for male children leads to significant social problems, medical sex selection can preclude the burden of a child with inherited disease. Nevertheless, avoiding non-medical sex selection remains a serious issue globally.

For the cffDNA samples with no Y-chromosome allele, the sex of all the infants had been determined through paternity testing by CE in our study. Thus, the cffDNA samples lacking Y-chromosome were known before NIPPT. However, in order to conduct a more rigorous study design, an internal control like Amelogenin locus should be included to distinguish DNA amplification failure or absence of Y-chromosome.

In summary, we established the proof-a-concept of Y-chromosome mini-STR-based NIPPT through NGS, which showed high accuracy in real cases and identified a mutation at DYS393. CPI and probability of paternity were calculated based on the local Y-STR haplotype frequencies. The discrimination power could be increased with the availability of more information regarding the haplotype. As an alternative of CE and SNP-based NGS, our approach can aid in familial searching, paternity exclusion and sex selection in forensic and medical applications.

## Supporting information

**S1 Table. Primers and repeat motif of 12 Y-chromosome mini-STR loci without barcoding sequences.**
(DOCX)

**S2 Table. Quantities of the 24 cffDNA extracts.**
(DOCX)

**S3 Table. Qualitative results for cffDNA Y-STR haplotypes and their matching to the infant and paternal genomic DNA haplotypes.**
(XLSX)

## Author Contributions

**Conceptualization:** Wenqian Song, Ming Liu.

**Data curation:** Wenqian Song, Nan Xiao, Weijian Yu, Ni Wang, Linnan Shao, Ying Duan, Mei Chen, Lingzi Pan, Yuexin Xia, Li Zhang.

**Formal analysis:** Wenqian Song, Shihang Zhou.

**Funding acquisition:** Wenqian Song.

**Methodology:** Wenqian Song, Nan Xiao, Shihang Zhou.

**Project administration:** Wenqian Song, Nan Xiao, Ni Wang, Linnan Shao, Ying Duan, Mei Chen, Lingzi Pan, Yuexin Xia, Li Zhang.

**Supervision:** Ming Liu.

**Validation:** Shihang Zhou, Ming Liu.

**Visualization:** Weijian Yu.

**Writing – original draft:** Wenqian Song.

**Writing – review & editing:** Wenqian Song, Ming Liu.

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
