## [Decision Letter · Decision Letter 0]

31 Aug 2021

PONE-D-21-09048

Non-invasive prenatal paternity testing by analysis of Y-chromosome mini-STR haplotype using next-generation sequencing

PLOS ONE

Dear Dr. Ming Liu,

Thank you for submitting your manuscript to PLOS ONE. After careful consideration, we feel that it has merit but does not fully meet PLOS ONE’s publication criteria as it currently stands. Therefore, we invite you to submit a revised version of the manuscript that addresses the points raised during the review process.

I would suggest that you read this following articles which will be of help to answer the reviewers' comments:  

1) Wang et al. STR polymorphisms of "forensic loci" in the northern Han Chinese population. Journal of Human Genetics July 2003, Volume 48, Issue 7, 337-341

2)  Tamaki et al. Microsatellite typing in a paternity case against a deceased man whose two brothers were available for testing.  Jpn J Leg Med 1996 50(2) 82-86

We look forward to receiving your revised manuscript.

Kind regards,

Wei Wang, M.D., Ph.D.

Academic Editor

PLOS ONE

Journal Requirements:

2. We note that you are reporting an analysis of a microarray, next-generation sequencing, or deep sequencing data set. PLOS requires that authors comply with field-specific standards for preparation, recording, and deposition of data in repositories appropriate to their field. Please upload these data to a stable, public repository (such as ArrayExpress, Gene Expression Omnibus (GEO), DNA Data Bank of Japan (DDBJ), NCBI GenBank, NCBI Sequence Read Archive, or EMBL Nucleotide Sequence Database (ENA)). In your revised cover letter, please provide the relevant accession numbers that may be used to access these data. For a full list of recommended repositories, see http://journals.plos.org/plosone/s/data-availability#loc-omics or http://journals.plos.org/plosone/s/data-availability#loc-sequencing.

Reviewers' comments:

Reviewer's Responses to Questions

**Comments to the Author**

1. Is the manuscript technically sound, and do the data support the conclusions?

Reviewer #1: Partly

Reviewer #2: Partly

Reviewer #3: Yes

2. Has the statistical analysis been performed appropriately and rigorously? 

Reviewer #1: Yes

Reviewer #2: Yes

Reviewer #3: Yes

3. Have the authors made all data underlying the findings in their manuscript fully available?

Reviewer #1: Yes

Reviewer #2: No

Reviewer #3: Yes

4. Is the manuscript presented in an intelligible fashion and written in standard English?

Reviewer #1: Yes

Reviewer #2: Yes

Reviewer #3: Yes

5. Review Comments to the Author

Reviewer #1: The manuscript describes a strategy for non-invasive prenatal paternity testing by analyzing the Y-chromosome. The method is useful and should be published following revisions (both major and minor) as indicated below (pages and line refer to the PDF format of the manuscript):

MAJOR COMMENTS

Materials and methods, under “DNA extraction”, page 4

1. For the following statement:

“The quantity and purity of the extracted DNA were determined by sodium dodecyl-sulphate polyacrylamide gel electrophoresis (SDS-PAGE) with Tanon 1600 Gel Imaging System (Shanghai Tanon Science & Technology, Shanghai, China) and NanoDrop 2000 spectrophotometry (Thermo Fisher Scientific, Waltham, MA, USA) respectively”:

provide reference or describe in more detail the methods used. In addition, provide information as to how much of the 60 μl extract was used for SDS-PAGE, for quantitation and for subsequent NGS library preparations.

2. Did the authors use any negative and/or reagent blank controls during DNA extractions and subsequent operations?

Materials and methods, under “Library preparation and NGS”, page 5

3. “DNA extracted from cell-free maternal plasma…”: Provide information as to how many μl of the cell-free DNA extract were used.

4. “…was amplified at the following 12 Y-chromosome mini-STR loci”: Provide the sequence information on the mini-STR primers that were designed for the present assay. In addition, usually barcoding in NGS is used for sample designation, therefore, it is essential that more details should be provided as to the NGS assay with respect to both the samples and the Y-STR loci assayed. More details are required to describe the developed NGS assay.

5. “The quantity and purity of the DNA libraries were assessed by SDS-PAGE.”: Provide reference for the method or more details.

Materials and methods, under “Calculation of paternity testing parameters”, page 6

6. “…the frequency of the son’s haplotype, µ is the mutation rate, n is the total number of Y-STR haplotypes,…”: Provide in this paragraph (i.e. in Materials and methods) the information regarding the use of a mutation rate of 0.00123 for locus DYS393 (indicated on page 8 of the MS under the Results section).

7. Also, another paragraph should be added to indicate that the CPIs were calculated based on 6 loci Y-haplotypes. In other words, the information provided at the beginning of page 10 of the manuscript: “Given the limited information on the Y-STR loci haplotypes in the Liaoning population, only 6 out of the 12 loci tested in our study could be used for calculating CPI and probability of paternity” should be provided in the Materials and Methods section.

Results, pages 6-8

8. The Results section must include (either in Table format or descriptive) information on the quantities of the 24 cell-free fetal DNA extracts as determined by the methods indicated in the Materials and Methods section.

9. As indicated before, a small note, preferably in the Materials and methods section, should indicate that the haplotype frequencies in Table 1 (page 8 of the manuscript) were estimated using the haplotypes for 6 Y-chromosomal loci.

Discussion

10. The inclusion of an internal control to verify that the lack of amplification is due to the absence of the Y-chromosome and not to experimental failure is an important aspect especially in real cases. Perhaps the Amelogenin locus, that has small amplicons could also be included in the assay. Authors to elaborate briefly on this (i.e. how to distinguish failure Vs absence of Y-chromosome) in the discussion section.

References

11. Please go over the references and make sure that authors, journals etc are cited correctly. For example Reference 14 should be corrected to read: Roewer L, Andersen MM, Ballantyne J, Butler JM, Caliebe M, Corach D, ME, Gusmão L, Hou Y, de Knijff P, Parson W, Prinz M, Schneider PM, Taylor D, Vennemann M, Willuweit S (2020) DNA commission of the International Society of Forensic Genetics (ISFG): Recommendations on the interpretation of Y-STR results in forensic analysis. Forensic Science International: Genetics 48: 102308

MINOR COMMENTS/SUGGESTIONS

Abstract

12. Suggestion to revise the sentence, page 1, to read: “The Y-chromosome mini-STR genotypes of all 14 male cffDNA were obtained…”

13. Suggestion to revise the sentence, page 2, to read: “The combined paternity index (CPI) and probability of paternity calculation was based on 6 loci Y-haplotype distributions of a local population.”

Introduction, pages 2

14. Suggestion to revise the sentence, page 2 (last s lines), to read: “Due to its uniparental inheritance, the results of Y-STR typing are interpreted on the basis of haplotype…”

Introduction, page 3

15. Suggestion to revise the sentence, page 3 (1st line), to read: “A major challenge in non-invasive prenatal paternity testing (NIPPT) is the detection of fetal-specific markers of the paternal genetic material…”

Materials and methods, page 4

16. Suggestion to revise the sentence, page 4 (under "DNA extraction", 1st line), to read: “A total of 5 mL peripheral blood was collected from the pregnant women in anticoagulant-treated tubes.”

17. Suggestion to revise the title to read: “Paternity testing by Capillary Electrophoresis (CE)”

Results, page 8

18. Suggestion to revise the title of the Table to read: “Table 1. The paternity testing parameters using cffDNA in the 14 cases with male fetuses”

Discussion, page 9

19. Suggestion to revise the sentence, page 9 (1st sentence - since the beginning of sentence spell out CE) to read: “Capillary Electrophoresis is the routine technique…”

20. Suggestion to revise the sentence, page 9 (lines 2-3), to read: “…genetic information that it can provide and process, high throughput…”

21. Suggestion to revise the sentence, page 9 (line 13), to read: “…millions of SNPs have to be analyzed due to low PIC (polymorphism information content) leading to statistics with lower power. In addition…”

22. Suggestion to revise the sentence, page 9 (lines 14-15), to read: “…SNPs in forensics is controversial since some SNPs may be present in the coding regions with bioethical implications stemming from confidentiality and privacy concerns since they may disclose sensitive information or are modified due to disease [29].”

23. Suggestion to revise the sentence, page 9 (lines 16-17), to read: “Therefore, given the limitations of CE and SNP-based NGS, mini-STR-based NGS with high testing capacity and high PIC is a better choice for NIPPT.”

24. Suggestion to revise the sentence, page 9 (last 2 lines), to read: “Due to strong linkage disequilibrium, the multiplying of single locus allele frequencies cannot be used for Y-STR CPI calculation.”

Discussion, page 10

25. It is recommended to add a small paragraph in Materials and Methods to describe the population that was used to calculate the haplotype frequencies in relation to what is written in the Discussion (page 10; lines 3-4): “In 7 of the 14 cases, the fetal haplotype were not detected in the 838 Liaoning haplotypes reported by Guo et al [22]. According to the ISFG algorithm [14]…”:

26. Suggestion to revise the sentence, page 10 (lines 19-21) to read: “...Y-chromosome mini-STR should be used for unwanted pregnancy as the result of sexual assault or for sex selection in case of gene defect and inherited disease.”

References, page 11

27. Suggestion to change the title to: “References” instead of “Reference”

Reviewer #2: Song et al. propose a proof-of-concept study in which they use massively parallel DNA sequencing to assign paternity to male fetuses in utero. The authors use CPI analysis and compare the alleged father's Y-chromosome STR haplotype to the fetus' haplotype, which is obtained noninvasively from the mother's blood plasma. The manuscript is well-written and the main ideas are clear.

Major comments:

While this study can be considered a valid contribution to science, my main concern is related to how the authors have interpreted their results. I’m not convinced that using only the Y-STR haplotype is sufficient to assign paternity with confidence for two reasons.

First, even though a CPI is calculated to show some degree of uncertainty to the paternity tests performed, we should keep in mind that the Y-STR haplotype represents only one locus in the genome. Thus, this Y-STR paternity analysis would be similar to accepting the use of only one autosomal locus as sufficient to determine paternity with confidence (which can also show greater than 99% CPIs depending on the locus, frequencies and alleles scored).

Second, unlike the autosomal loci, the Y-chromosome is inherited from father to son without recombination. This mode of inheritance creates the potential for several men in the population to share the exact same Y-STR profile (i.e. they belong to the same lineage). For instance, if the paternal grandfather of one of those fetuses had four brothers and each of those brothers (including the grandfather himself) had four sons, and each of those 16 men had two sons, we would have, in this family alone, 52 individuals that would, in theory, match the fetus’ Y-STR profile. This isn’t the case when analyzing autosomal STR profiles (with at least a dozen loci each) due to recombination. In other words, only identical twins are expected to share the same STR profile in the population while several men can (and do) share the same Y-STR haplotype, creating the real potential of false positive results.

For these reasons, unlike it’s implied in the title and throughout manuscript, Y-STR haplotype results alone should not be used to confirm paternity. Instead, it could be used to “exclude” or “not exclude” a male from being a potential father of a fetus. Especially in the forensic setting, where paternity tests can have broad impacts on people's lives, conclusions should always be confirmed with more reliable methods such as genotyping multiple autosomal STR loci.

Minor comments:

1) In the Abstract, where it’s said “The cffDNA genotype was validated by the paternal genotype” I believe the word “genotype” should be replaced by “haplotype” as this sentence seems to be related to the Y-chromosome. I also suggest this wording is checked throughout the manuscript.

2) At the end of the Abstract and the Discussion, the authors mention “sex selection”. This is a very controversial subject due to its ethical implications. And as the authors mention, such practice is forbidden in many countries. Should this potential application be promoted or even discussed in this publication?

3) Materials and Methods (Sample collection): I suggest rewording the first sentence this way: “Peripheral blood samples were collected from 24 pregnant women undergoing prenatal tests, and their male partners, at the Dalian Blood Centre from April 2018 to December 2019”. Dalian Blood *Centre* or Dalian Blood *Center*?

4) Materials and Methods (DNA extraction/library preparation and NGS): Assessing the quantity and purity of DNA extracts and library preparations with SDS-PAGE doesn’t seem to be standard practice. Can you describe a little more how this is done?

5) Figure 2 (flowchart): the boxes “Infant STR genotypes” and “Paternal STR genotypes” appear to be swapped.

Reviewer #3: The study applied a 12 Y-STR multiplex to pre-natal testing of foetal DNA in maternal circulation. Standard statistical paternity testing regimes were applied to confirm paternity. A novel aspect of the work is the use of NGS sequencing to genotype the STRs. However, no primer details or sequence output analysis is given - the Results section is extremely thin, as if the authors assume that the readership would not be interested in how well the NGS assay for these 12 Y-STRs worked with such low-level DNA input. The authors do not discuss their choice of Y loci - why select these 12?

No description is given for the extent to which sequence variation could be used and potentially was of value in the paternity analyses made. This was because normal CE was used to type the matched father's DNA, and NGS for the maternal samples, whereas, it would be potentially better to establish how much sequence variation could benefit the paternity statistical analyses in each case. Therefore, the authors lost every opportunity to report in this paper the sensitivity of the system they have developed in terms of sequence coverage and expanded identification discrimination from sequence variants in the chosen loci.

As such, the work was made scientifically, but there is little of novelty or impact in the study (and I realise the paper cannot be rejected on these grounds alone).

Figures 1 and 2 are largely redundant.

An initial text slip suggested an English review would be beneficial, but in fact, the standard of the rest of the paper is fine:

Our proof-a-concept study demonstrated that Y-chromosome mini-STR can be used > Our proof-of-concept study demonstrated that Y-chromosome mini-STRs can be used.

6. PLOS authors have the option to publish the peer review history of their article (what does this mean?). If published, this will include your full peer review and any attached files.

Reviewer #1: No

Reviewer #2: No

Reviewer #3: **Yes: **Christopher Phillips

---

## [Author Response · Author response to Decision Letter 0]

10 Oct 2021

Dear Dr. Wang,

Thank you very much for giving us an opportunity to revise our manuscript entitled “Non-invasive prenatal paternity testing by analysis of Y-chromosome mini-STR haplotype using next-generation sequencing”. We are supposed to upload the NGS data, including Metadata spreadsheet, Processed data and Raw data to GEO. But more time is needed for us to get Processed data from Microread Genetics, which is the NGS service supplier for us, due to the laboratory technologist who was in charge of our study resigned from Microread Genetics. We apologize that we cannot upload the NGS data to GEO and provide the relevant accession numbers before October 11. May we provide them later?

The comments from the reviewers have been very helpful in improving our paper and designing new experiments. The revised portions in the manuscript are marked in yellow. The responses to the reviewers’ comments are as follows.

Reviewer #1: The manuscript describes a strategy for non-invasive prenatal paternity testing by analyzing the Y-chromosome. The method is useful and should be published following revisions (both major and minor) as indicated below (pages and line refer to the PDF format of the manuscript):

MAJOR COMMENTS

Materials and methods, under “DNA extraction”, page 4

1. For the following statement:

“The quantity and purity of the extracted DNA were determined by sodium dodecyl-sulphate polyacrylamide gel electrophoresis (SDS-PAGE) with Tanon 1600 Gel Imaging System (Shanghai Tanon Science & Technology, Shanghai, China) and NanoDrop 2000 spectrophotometry (Thermo Fisher Scientific, Waltham, MA, USA) respectively”:

provide reference or describe in more detail the methods used. In addition, provide information as to how much of the 60 μl extract was used for SDS-PAGE, for quantitation and for subsequent NGS library preparations.

Answer: “Presence of contaminants such as RNA and proteins … was estimated on the basis of a molecular weight marker band.” has been added under “DNA extraction”.

“1 µL” has been added in the “The quantity and purity of the 1µL extracted DNA were determined by sodium dodecyl-sulphate polyacrylamide gel electrophoresis (SDS-PAGE)…”.

2. Did the authors use any negative and/or reagent blank controls during DNA extractions and subsequent operations?

Materials and methods, under “Library preparation and NGS”, page 5

Answer: “The 2800M Control DNA (Promega, Madison, WI, USA) was used as the positive control, and nuclease-free water was used as the PCR blank.” has been added under “DNA extraction”

“The 2800M Control DNA was used to construct the library.” has been added under “Library preparation and NGS”.

3. “DNA extracted from cell-free maternal plasma…”: Provide information as to how many μl of the cell-free DNA extract were used.

Answer: It has been changed to “1µL of DNA extract from cell-free maternal plasma …”

4. “…was amplified at the following 12 Y-chromosome mini-STR loci”: Provide the sequence information on the mini-STR primers that were designed for the present assay. In addition, usually barcoding in NGS is used for sample designation, therefore, it is essential that more details should be provided as to the NGS assay with respect to both the samples and the Y-STR loci assayed. More details are required to describe the developed NGS assay.

Answer: The sequence information of mini-STR primers was attached in the S1 Table. 

More details about NGS assay have been added in Figure 1. 

In the manuscript, “which mainly includes addition of specialized adapters and indexes to both ends of barcoding sequences (Fig 1) … 15 cycles of 95 °C for 30 s, 60 °C for 30 s, 72 °C for 30 s; 72 °C for 5 min.” has been added under “Library preparation and NGS”.

5. “The quantity and purity of the DNA libraries were assessed by SDS-PAGE.”: Provide reference for the method or more details.

Answer: The DNA libraries were constructed by two-step PCR and purified using MagPure A3 XP beads. We assessed the quantity and purity of libraries before and after the purification by SDS-PAGE and Qubit fluorometer respectively. For more details, “Each DNA library was normalized to 2nM and pooled in equal volumes. Finally, the DNA libraries were diluted to the concentration of 10pM and…” has been added under “Library preparation and NGS”.

Materials and methods, under “Calculation of paternity testing parameters”, page 6

6. “…the frequency of the son’s haplotype, µ is the mutation rate, n is the total number of Y-STR haplotypes,…”: Provide in this paragraph (i.e. in Materials and methods) the information regarding the use of a mutation rate of 0.00123 for locus DYS393 (indicated on page 8 of the MS under the Results section).

Answer: “For the case with a mutation, the mutation rate µ can be obtained from YHRD. According to equation (1), m=1 was used, and CPI=(1/2µ)/fs.” has been added under “Calculation of paternity testing parameters”.

7. Also, another paragraph should be added to indicate that the CPIs were calculated based on 6 loci Y-haplotypes. In other words, the information provided at the beginning of page 10 of the manuscript: “Given the limited information on the Y-STR loci haplotypes in the Liaoning population, only 6 out of the 12 loci tested in our study could be used for calculating CPI and probability of paternity” should be provided in the Materials and Methods section.

Answer: The paragraphs “The CPI and probability of paternity calculation rely on the haplotype distribution in the local population … were used in cases without mutation, CPI=1/fs.” has been added under “Calculation of paternity testing parameters”.

Results, pages 6-8

8. The Results section must include (either in Table format or descriptive) information on the quantities of the 24 cell-free fetal DNA extracts as determined by the methods indicated in the Materials and Methods section.

Answer: “The quantities of the 24 cell-free fetal DNA extracts were shown in S2 Table.” has been added under “STR genotype-match analysis between cffDNA and paternal genomic DNA”.

9. As indicated before, a small note, preferably in the Materials and methods section, should indicate that the haplotype frequencies in Table 1 (page 8 of the manuscript) were estimated using the haplotypes for 6 Y-chromosomal loci.

Answer: “Given the limited information on the Y-STR loci haplotypes in the Liaoning population, only 6 out of the 12 loci tested in our study could be used for estimating haplotype frequencies, calculating CPI and probability of paternity.” has been added under “Calculation of paternity testing parameters”.

Discussion

10. The inclusion of an internal control to verify that the lack of amplification is due to the absence of the Y-chromosome and not to experimental failure is an important aspect especially in real cases. Perhaps the Amelogenin locus, that has small amplicons could also be included in the assay. Authors to elaborate briefly on this (i.e. how to distinguish failure Vs absence of Y-chromosome) in the discussion section.

Answer: “For the cffDNA samples with no Y-chromosome allele … distinguish DNA amplification failure or absence of Y-chromosome.” has been added under “Discussion”.

References

11. Please go over the references and make sure that authors, journals etc are cited correctly. For example Reference 14 should be corrected to read: Roewer L, Andersen MM, Ballantyne J, Butler JM, Caliebe M, Corach D, ME, Gusmão L, Hou Y, de Knijff P, Parson W, Prinz M, Schneider PM, Taylor D, Vennemann M, Willuweit S (2020) DNA commission of the International Society of Forensic Genetics (ISFG): Recommendations on the interpretation of Y-STR results in forensic analysis. Forensic Science International: Genetics 48: 102308

Answer: The mistakes in the references have been corrected and highlighted under “Reference”.

MINOR COMMENTS/SUGGESTIONS

Abstract

12. Suggestion to revise the sentence, page 1, to read: “The Y-chromosome mini-STR genotypes of all 14 male cffDNA were obtained…”

Answer: It has been revised in the manuscript.

13. Suggestion to revise the sentence, page 2, to read: “The combined paternity index (CPI) and probability of paternity calculation was based on 6 loci Y-haplotype distributions of a local population.”

Answer: It has been revised in the manuscript.

Introduction, pages 2

14. Suggestion to revise the sentence, page 2 (last s lines), to read: “Due to its uniparental inheritance, the results of Y-STR typing are interpreted on the basis of haplotype…”

Answer: It has been revised in the manuscript.

Introduction, page 3

15. Suggestion to revise the sentence, page 3 (1st line), to read: “A major challenge in non-invasive prenatal paternity testing (NIPPT) is the detection of fetal-specific markers of the paternal genetic material…”

Answer: It has been revised in the manuscript.

Materials and methods, page 4

16. Suggestion to revise the sentence, page 4 (under "DNA extraction", 1st line), to read: “A total of 5 mL peripheral blood was collected from the pregnant women in anticoagulant-treated tubes.”

Answer: It has been revised in the manuscript.

17. Suggestion to revise the title to read: “Paternity testing by Capillary Electrophoresis (CE)”

Answer: It has been revised in the manuscript.

Results, page 8

18. Suggestion to revise the title of the Table to read: “Table 1. The paternity testing parameters using cffDNA in the 14 cases with male fetuses”

Answer: It has been revised in the manuscript.

Discussion, page 9

19. Suggestion to revise the sentence, page 9 (1st sentence - since the beginning of sentence spell out CE) to read: “Capillary Electrophoresis is the routine technique…”

Answer: It has been revised in the manuscript.

20. Suggestion to revise the sentence, page 9 (lines 2-3), to read: “…genetic information that it can provide and process, high throughput…”

Answer: It has been revised in the manuscript.

21. Suggestion to revise the sentence, page 9 (line 13), to read: “…millions of SNPs have to be analyzed due to low PIC (polymorphism information content) leading to statistics with lower power. In addition…”

Answer: It has been revised in the manuscript.

22. Suggestion to revise the sentence, page 9 (lines 14-15), to read: “…SNPs in forensics is controversial since some SNPs may be present in the coding regions with bioethical implications stemming from confidentiality and privacy concerns since they may disclose sensitive information or are modified due to disease [29].”

Answer: It has been revised in the manuscript.

23. Suggestion to revise the sentence, page 9 (lines 16-17), to read: “Therefore, given the limitations of CE and SNP-based NGS, mini-STR-based NGS with high testing capacity and high PIC is a better choice for NIPPT.”

Answer: It has been revised in the manuscript.

24. Suggestion to revise the sentence, page 9 (last 2 lines), to read: “Due to strong linkage disequilibrium, the multiplying of single locus allele frequencies cannot be used for Y-STR CPI calculation.”

Answer: It has been revised in the manuscript.

Discussion, page 10

25. It is recommended to add a small paragraph in Materials and Methods to describe the population that was used to calculate the haplotype frequencies in relation to what is written in the Discussion (page 10; lines 3-4): “In 7 of the 14 cases, the fetal haplotype were not detected in the 838 Liaoning haplotypes reported by Guo et al [22]. According to the ISFG algorithm [14]…”:

Answer: “For example, one of 14 Y-haplotypes with 6 loci in our study matches two of the 838 Liaoning haplotypes available at the YHRD, that results in a haplotype frequency of 0.003521 ((1+2)/(14+838)). Then, CPI and probability of paternity could be calculated based on this haplotype frequency.” has been added under “Calculation of paternity testing parameters”.

26. Suggestion to revise the sentence, page 10 (lines 19-21) to read: “...Y-chromosome mini-STR should be used for unwanted pregnancy as the result of sexual assault or for sex selection in case of gene defect and inherited disease.”

Answer: It has been revised in the manuscript.

References, page 11

27. Suggestion to change the title to: “References” instead of “Reference”

Answer: It has been revised in the manuscript.

Reviewer #2: Song et al. propose a proof-of-concept study in which they use massively parallel DNA sequencing to assign paternity to male fetuses in utero. The authors use CPI analysis and compare the alleged father's Y-chromosome STR haplotype to the fetus' haplotype, which is obtained noninvasively from the mother's blood plasma. The manuscript is well-written and the main ideas are clear.

Major comments:

While this study can be considered a valid contribution to science, my main concern is related to how the authors have interpreted their results. I’m not convinced that using only the Y-STR haplotype is sufficient to assign paternity with confidence for two reasons.

First, even though a CPI is calculated to show some degree of uncertainty to the paternity tests performed, we should keep in mind that the Y-STR haplotype represents only one locus in the genome. Thus, this Y-STR paternity analysis would be similar to accepting the use of only one autosomal locus as sufficient to determine paternity with confidence (which can also show greater than 99% CPIs depending on the locus, frequencies and alleles scored).

Second, unlike the autosomal loci, the Y-chromosome is inherited from father to son without recombination. This mode of inheritance creates the potential for several men in the population to share the exact same Y-STR profile (i.e. they belong to the same lineage). For instance, if the paternal grandfather of one of those fetuses had four brothers and each of those brothers (including the grandfather himself) had four sons, and each of those 16 men had two sons, we would have, in this family alone, 52 individuals that would, in theory, match the fetus’ Y-STR profile. This isn’t the case when analyzing autosomal STR profiles (with at least a dozen loci each) due to recombination. In other words, only identical twins are expected to share the same STR profile in the population while several men can (and do) share the same Y-STR haplotype, creating the real potential of false positive results.

For these reasons, unlike it’s implied in the title and throughout manuscript, Y-STR haplotype results alone should not be used to confirm paternity. Instead, it could be used to “exclude” or “not exclude” a male from being a potential father of a fetus. Especially in the forensic setting, where paternity tests can have broad impacts on people's lives, conclusions should always be confirmed with more reliable methods such as genotyping multiple autosomal STR loci.

Minor comments:

1) In the Abstract, where it’s said “The cffDNA genotype was validated by the paternal genotype” I believe the word “genotype” should be replaced by “haplotype” as this sentence seems to be related to the Y-chromosome. I also suggest this wording is checked throughout the manuscript.

Answer: Thanks for your suggestion. All the words“genotype” have been revised in the manuscript.

2) At the end of the Abstract and the Discussion, the authors mention “sex selection”. This is a very controversial subject due to its ethical implications. And as the authors mention, such practice is forbidden in many countries. Should this potential application be promoted or even discussed in this publication?

Answer: We have discussed more about sex selection in the manuscript. “Sex selection can be a double-edged sword … avoiding non-medical sex selection remains a serious issue globally.” has been added under “Discussion”.

3) Materials and Methods (Sample collection): I suggest rewording the first sentence this way: “Peripheral blood samples were collected from 24 pregnant women undergoing prenatal tests, and their male partners, at the Dalian Blood Centre from April 2018 to December 2019”. Dalian Blood *Centre* or Dalian Blood *Center*?

Answer: The sentence has been revised in the manuscript. I think both “Centre” and “Center” are okay.

4) Materials and Methods (DNA extraction/library preparation and NGS): Assessing the quantity and purity of DNA extracts and library preparations with SDS-PAGE doesn’t seem to be standard practice. Can you describe a little more how this is done?

Answer: “Presence of contaminants such as RNA and proteins … was estimated on the basis of a molecular weight marker band.” has been added under “DNA extraction”.

5) Figure 2 (flowchart): the boxes “Infant STR genotypes” and “Paternal STR genotypes” appear to be swapped.

Answer: The revised figure 2 has been attached.

Reviewer #3: The study applied a 12 Y-STR multiplex to pre-natal testing of foetal DNA in maternal circulation. Standard statistical paternity testing regimes were applied to confirm paternity. A novel aspect of the work is the use of NGS sequencing to genotype the STRs. However, no primer details or sequence output analysis is given - the Results section is extremely thin, as if the authors assume that the readership would not be interested in how well the NGS assay for these 12 Y-STRs worked with such low-level DNA input. The authors do not discuss their choice of Y loci - why select these 12?

Answer: Thanks for your suggestion. The primer sequences have been provided in S1 Table. 

We chose these 12 Y loci due to their superior multiplex amplification compared to the others. 

No description is given for the extent to which sequence variation could be used and potentially was of value in the paternity analyses made. This was because normal CE was used to type the matched father's DNA, and NGS for the maternal samples, whereas, it would be potentially better to establish how much sequence variation could benefit the paternity statistical analyses in each case. Therefore, the authors lost every opportunity to report in this paper the sensitivity of the system they have developed in terms of sequence coverage and expanded identification discrimination from sequence variants in the chosen loci.

Answer: We appreciate your comment about extending the study to sequence variation, which will be dealt in a subsequent study. 

As such, the work was made scientifically, but there is little of novelty or impact in the study (and I realise the paper cannot be rejected on these grounds alone).

Figures 1 and 2 are largely redundant.

Answer: Figures 1 and 2 were exported from Adobe Illustrator Artwork and Microsoft Visio with the highest resolution.

An initial text slip suggested an English review would be beneficial, but in fact, the standard of the rest of the paper is fine:

Our proof-a-concept study demonstrated that Y-chromosome mini-STR can be used > Our proof-of-concept study demonstrated that Y-chromosome mini-STRs can be used.

Answer: The word “proof-a-concept” has been changed to “proof-of-concept” under “Abstract”.

---

## [Decision Letter · Decision Letter 1]

15 Nov 2021

PONE-D-21-09048R1Non-invasive prenatal paternity testing by analysis of Y-chromosome mini-STR haplotype using next-generation sequencingPLOS ONE

Dear Dr. Liu,

Thank you for submitting your manuscript to PLOS ONE. After careful consideration, we feel that it has merit but does not fully meet PLOS ONE’s publication criteria as it currently stands. Therefore, we invite you to submit a revised version of the manuscript that addresses the points raised during the review process. Please submit your revised manuscript by Dec 30 2021 11:59PM. If you will need more time than this to complete your revisions, please reply to this message or contact the journal office at plosone@plos.org. Please include the following items when submitting your revised manuscript:A rebuttal letter that responds to each point raised by the academic editor and reviewer(s). You should upload this letter as a separate file labeled 'Response to Reviewers'.A marked-up copy of your manuscript that highlights changes made to the original version. You should upload this as a separate file labeled 'Revised Manuscript with Track Changes'.An unmarked version of your revised paper without tracked changes. You should upload this as a separate file labeled 'Manuscript'.If applicable, we recommend that you deposit your laboratory protocols in protocols.io to enhance the reproducibility of your results. Protocols.io assigns your protocol its own identifier (DOI) so that it can be cited independently in the future. For instructions see: https://journals.plos.org/plosone/s/submission-guidelines#loc-laboratory-protocols. Additionally, PLOS ONE offers an option for publishing peer-reviewed Lab Protocol articles, which describe protocols hosted on protocols.io. Read more information on sharing protocols at https://plos.org/protocols?utm_medium=editorial-email&utm_source=authorletters&utm_campaign=protocols.

We look forward to receiving your revised manuscript.

Kind regards,

Kelvin Yuen-Kwong CHAN, Ph.D.

Academic Editor

PLOS ONE

Journal Requirements:

Reviewers' comments:

Reviewer's Responses to Questions

**Comments to the Author**

1. If the authors have adequately addressed your comments raised in a previous round of review and you feel that this manuscript is now acceptable for publication, you may indicate that here to bypass the “Comments to the Author” section, enter your conflict of interest statement in the “Confidential to Editor” section, and submit your "Accept" recommendation.

Reviewer #1: All comments have been addressed

Reviewer #2: (No Response)

Reviewer #3: All comments have been addressed

2. Is the manuscript technically sound, and do the data support the conclusions?

Reviewer #1: Yes

Reviewer #2: Partly

Reviewer #3: Yes

3. Has the statistical analysis been performed appropriately and rigorously? 

Reviewer #1: Yes

Reviewer #2: Yes

Reviewer #3: Yes

4. Have the authors made all data underlying the findings in their manuscript fully available?

Reviewer #1: Yes

Reviewer #2: No

Reviewer #3: Yes

5. Is the manuscript presented in an intelligible fashion and written in standard English?

Reviewer #1: Yes

Reviewer #2: Yes

Reviewer #3: Yes

6. Review Comments to the Author

Reviewer #1: (No Response)

Reviewer #2: Dear colleagues:

In this current version, I believe that Song et al. have addressed most of the criticisms raised by the reviewers in the first submission. However, I'm still VERY CONCERNED about the fact that the authors are proposing a paternity test based exclusivelly on Y-STR haplotypes. In fact, my major comments, which I copy again below, were not responded by the authors.

I would like to stress that unlike autosomal STR genotypes, IDENTICAL Y-STR haplotypes are found in the population because the Y-chromosome is passed on from father to son without recombination. Thus, I strongly suggest that the authors at least acknowledge in the manuscript the possibility of false-positive results due to the inheriteance mode of Y-STR haplotypes. And ideally, the authors could also recommend that the Y-STR results must be confirmed with autosomal STR markers when a sample can be safely collected from the fetus/baby (e.g. after birth or miscarriage).

FROM PREVIOUS REVIEW:

First, even though a CPI is calculated to show some degree of uncertainty to the paternity tests performed, we should keep in mind that the Y-STR haplotype represents only one locus in the genome. Thus, this Y-STR paternity analysis would be similar to accepting the use of only one autosomal locus as sufficient to determine paternity with confidence (which can also show greater than 99% CPIs depending on the locus, frequencies and alleles scored).

Second, unlike the autosomal loci, the Y-chromosome is inherited from father to son without recombination. This mode of inheritance creates the potential for several men in the population to share the exact same Y-STR profile (i.e. they belong to the same lineage). For instance, if the paternal grandfather of one of those fetuses had four brothers and each of those brothers (including the grandfather himself) had four sons, and each of those 16 men had two sons, we would have, in this family alone, 52 individuals that would, in theory, match the fetus’ Y-STR profile. This isn’t the case when analyzing autosomal STR profiles (with at least a dozen loci each) due to recombination. In other words, only identical twins are expected to share the same STR profile in the population while several men can (and do) share the same Y-STR haplotype, creating the real potential of false positive results.

For these reasons, unlike it’s implied in the title and throughout manuscript, Y-STR haplotype results alone should not be used to confirm paternity. Instead, it could be used to “exclude” or “not exclude” a male from being a potential father of a fetus. Especially in the forensic setting, where paternity tests can have broad impacts on people's lives, conclusions should always be confirmed with more reliable methods such as genotyping multiple autosomal STR loci.

Reviewer #3: All required clarifications and explanations in the text have been made. The description of the figures as 'redundant' was meant to signify that they served no purpose and could be excluded - but their retention is not a problem. I understand the underlying NGS data will be made available in due course, even thought the authors cannot meet this obligation at the moment.

7. PLOS authors have the option to publish the peer review history of their article (what does this mean?). If published, this will include your full peer review and any attached files.

Reviewer #1: No

Reviewer #2: No

Reviewer #3: **Yes: **Christopher Phillips

---

## [Author Response · Author response to Decision Letter 1]

26 Nov 2021

Dear Dr. Chan,

Thank you for giving us another chance to revise our manuscript entitled “Non-invasive prenatal paternity testing by analysis of Y-chromosome mini-STR haplotype using next-generation sequencing”. We appreciate the constructive criticism from the reviewers, and have made the necessary changes in the rebuttal letter and manuscript. The revised portion in the manuscript is marked in yellow. The responses to the reviewers’ comments are as follows.

Reviewer #2: Dear colleagues:

In this current version, I believe that Song et al. have addressed most of the criticisms raised by the reviewers in the first submission. However, I'm still VERY CONCERNED about the fact that the authors are proposing a paternity test based exclusivelly on Y-STR haplotypes. In fact, my major comments, which I copy again below, were not responded by the authors.

I would like to stress that unlike autosomal STR genotypes, IDENTICAL Y-STR haplotypes are found in the population because the Y-chromosome is passed on from father to son without recombination. Thus, I strongly suggest that the authors at least acknowledge in the manuscript the possibility of false-positive results due to the inheriteance mode of Y-STR haplotypes. And ideally, the authors could also recommend that the Y-STR results must be confirmed with autosomal STR markers when a sample can be safely collected from the fetus/baby (e.g. after birth or miscarriage).

FROM PREVIOUS REVIEW:

First, even though a CPI is calculated to show some degree of uncertainty to the paternity tests performed, we should keep in mind that the Y-STR haplotype represents only one locus in the genome. Thus, this Y-STR paternity analysis would be similar to accepting the use of only one autosomal locus as sufficient to determine paternity with confidence (which can also show greater than 99% CPIs depending on the locus, frequencies and alleles scored).

Second, unlike the autosomal loci, the Y-chromosome is inherited from father to son without recombination. This mode of inheritance creates the potential for several men in the population to share the exact same Y-STR profile (i.e. they belong to the same lineage). For instance, if the paternal grandfather of one of those fetuses had four brothers and each of those brothers (including the grandfather himself) had four sons, and each of those 16 men had two sons, we would have, in this family alone, 52 individuals that would, in theory, match the fetus’ Y-STR profile. This isn’t the case when analyzing autosomal STR profiles (with at least a dozen loci each) due to recombination. In other words, only identical twins are expected to share the same STR profile in the population while several men can (and do) share the same Y-STR haplotype, creating the real potential of false positive results.

For these reasons, unlike it’s implied in the title and throughout manuscript, Y-STR haplotype results alone should not be used to confirm paternity. Instead, it could be used to “exclude” or “not exclude” a male from being a potential father of a fetus. Especially in the forensic setting, where paternity tests can have broad impacts on people's lives, conclusions should always be confirmed with more reliable methods such as genotyping multiple autosomal STR loci.

Answer: We apologize for not responding your major comments last time. “Unlike the autosomal loci … autosomal STR genotyping to determine paternity.” has been added in the manuscript.

Reviewer #3: All required clarifications and explanations in the text have been made. The description of the figures as 'redundant' was meant to signify that they served no purpose and could be excluded - but their retention is not a problem. I understand the underlying NGS data will be made available in due course, even thought the authors cannot meet this obligation at the moment.

Answer: Thanks for the suggestion. If possible, I hope the description of the figures can be retained. In addition, NGS data has been uploaded to GEO, and the relevant accession number is GSE186434.

---

## [Decision Letter · Decision Letter 2]

21 Mar 2022

Non-invasive prenatal paternity testing by analysis of Y-chromosome mini-STR haplotype using next-generation sequencing

PONE-D-21-09048R2

Dear Dr. Liu,

We’re pleased to inform you that your manuscript has been judged scientifically suitable for publication and will be formally accepted for publication once it meets all outstanding technical requirements.

Please kindly take note the following comments from Reviewer #2 when revising your manuscript:

1) Abstract - This sentence my be missing a period: "The cffDNA haplotype was validated by the paternal haplotype The paternity testing parameters were attributed to each case quantitatively."

2) Figure 2 (and throughout the manuscript) - In some cases it's unclear when the authors are talking about Y-STR HAPLOTYPES or AUTOSOMAL STR GENOTYPES since they did both (e.g. Fig. 2). To better distinguish those, I suggest that when talking about Y-STRs the word "haplotype" is used and when talking about autosomal STRs, the word "genotype" or the phrase "autosomal genotype" is used.

Kind regards,

Kelvin Yuen-Kwong CHAN, Ph.D.

Academic Editor

PLOS ONE

Additional Editor Comments (optional):

Reviewers' comments:

Reviewer's Responses to Questions

**Comments to the Author**

1. If the authors have adequately addressed your comments raised in a previous round of review and you feel that this manuscript is now acceptable for publication, you may indicate that here to bypass the “Comments to the Author” section, enter your conflict of interest statement in the “Confidential to Editor” section, and submit your "Accept" recommendation.

Reviewer #2: All comments have been addressed

2. Is the manuscript technically sound, and do the data support the conclusions?

Reviewer #2: Yes

3. Has the statistical analysis been performed appropriately and rigorously? 

Reviewer #2: Yes

4. Have the authors made all data underlying the findings in their manuscript fully available?

Reviewer #2: Yes

5. Is the manuscript presented in an intelligible fashion and written in standard English?

Reviewer #2: Yes

6. Review Comments to the Author

Reviewer #2: Abstract - This sentence my be missing a period: "The cffDNA haplotype was validated by the paternal haplotype The paternity testing parameters were attributed to each case quantitatively."

Figure 2 (and throughout the manuscript) - In some cases it's unclear when the authors are talking about Y-STR HAPLOTYPES or AUTOSOMAL STR GENOTYPES since they did both (e.g. Fig. 2). To better distinguish those, I suggest that when talking about Y-STRs the word "haplotype" is used and when talking about autosomal STRs, the word "genotype" or the phrase "autosomal genotype" is used.

7. PLOS authors have the option to publish the peer review history of their article (what does this mean?). If published, this will include your full peer review and any attached files.

Reviewer #2: No

---

## [Editor Report · Acceptance letter]

23 Mar 2022

PONE-D-21-09048R2 

Non-invasive prenatal paternity testing by analysis of Y-chromosome mini-STR haplotype using next-generation sequencing 

Dear Dr. Liu:

I'm pleased to inform you that your manuscript has been deemed suitable for publication in PLOS ONE. Congratulations! Your manuscript is now with our production department. 

Kind regards, 

on behalf of

Dr. Kelvin Yuen-Kwong CHAN 

Academic Editor

PLOS ONE